# Dual-Band Light Absorption Enhancement in Hyperbolic Rectangular Array

**Honglong Qi** [1,2], **Tian Sang** [1,2,*], **La Wang** [1,2], **Xin Yin** [1,2], **Jicheng Wang** [1,2] and **Yueke Wang** [1,2]

[1] Department of Photoelectric Information Science and Engineering, School of Science, Jiangnan University, Wuxi 214122, China; 18860478830@163.com (H.Q.); lawang1018@163.com (L.W.); xinyin0202@163.com (X.Y.); jcwang@jiangnan.edu.cn (J.W.); ykwang@jiangnan.edu.cn (Y.W.)

[2] Jiangsu Provincial Research Center of Light Industrial Optoelectronic Engineering and Technology, Jiangnan University, Wuxi 214122, China

* Correspondence: sangt@jiangnan.edu.cn

**Abstract:** The effect of dual-band light absorption enhancement in a hyperbolic rectangular array (HRA) is presented. The enhanced light absorption of the HRA results from the propagating surface plasmon (PSP) resonance, and a dual-band absorption with low and flat sideband level can be realized. The impedance theory is used to evaluate the absorption properties of the HRA, and shows that the input impedances of the HRA varied abruptly around the absorption bands to meet the impedance matching. The absorption spectra of the HRA can be estimated using the effective medium theory (EMT), and its accuracy can be improved as the number of film stacks is increased. The dual-band absorptions of the HRA are very robust to the variations of the width and the number of film stack. Potential application in refractive index sensing can be achieved by utilizing the two absorption bands.

**Keywords:** dual-band absorption enhancement; hyperbolic metamaterials; hyperbolic rectangular arrays; input impedances; refractive index sensing

## 1. Introduction

With recent advances in nanoscale fabrication of metal-dielectric multilayers and arrays of rods, hyperbolic metamaterials (HMMs) have attracted a great deal of interest because of their unusual optical features [1,2]. HMMs are characterized by the diagonal permittivity tensor with the principal components being of opposite signs. The corresponding dispersion relation permits wavevectors that lie within a hyperbolic isofrequency surface [3,4]. This special hyperboloid isofrequency surface allows HMMs to achieve exotic physical properties and functionality unattainable with naturally occurring materials.

In recent years, HMMs have been widely studied in many fields, such as sub-wavelength imaging [5], negative refraction [6,7], super-Planckian thermal emission [8], biosensing [9,10] and nanolasers [11]. From a practical point of view, a strong light absorption of HMMs is essential for various potential applications such as photoelectric detection and solar cells. It has been shown that HMMs can be used to enhance light absorption due to their high-k modes and divergent photonic density of states [4,12]. To date, many efforts have been made to improve light absorption of HMMs with the absorption wavelengths covering the microwave [13], terahertz [14–16], infrared [17–22] and visible wavelength regions [23–27]. In real applications, selective absorption enhancement is highly desired in fields such as thermal radiation tailoring [28], sensitive detection [29] and sensing [30]. However, previous studies of HMMs mainly focused on light absorption enhancement in broad spectral bands [13–19,21–27], and there are only a few studies on selective absorption enhancement of HMMs.

In this study, dual-band light absorption enhancement of HMMs is proposed by using a hyperbolic rectangular array (HRA). The enhanced light absorption of the HRA results from the propagating surface plasmon (PSP) resonance, and double absorption bands with peak absorption of ~95% in the visible-to-near-infrared region can be obtained. The evolution of the selective absorption properties of the HRA are investigated by using the impedance theory and the effective medium theory (EMT). The dual-band absorption is very robust to the variations of the width and the number of film stack. When these types of HMMs are used in refractive index sensing, the HRA shows good sensor performance due to its stable absorption enhancement.

## 2. Design Principles

Figure 1a shows the schematic diagram of the HRA, which consists of alternating silver (Ag) and silicon dioxide (SiO$_2$) films on top of an Ag substrate. The thickness of the Ag film ($t_m$) is 10 nm, and its dielectric constant ($\varepsilon_m$) is described by a multioscillator (Lorentz-Drude model) with the parameters given by Rakic et al. [31]. The thickness of SiO$_2$ ($t_d$) is 35 nm and its dielectric constant ($\varepsilon_d$) is 2.16. $f = t_m/(t_d + t_m) = 0.22$ is the filling ratio of the Ag film. The number of film stack of alternating Ag/SiO$_2$ pairs ($N$) is 8, and the width of film stack is $W$. $P$ and $D$ are the period and depth of the HRA, respectively. Subwavelength structure ($P < \lambda$) is required so as to achieve high absorption efficiency. The Ag substrate is chosen to be optically thick enough ($t_s = 100$ nm) to prevent light transmission, thus light absorption (A) can be reduced as A = 1 − R, where R is the reflection of the HRA. In simulation, the 2-dimensional finite-difference time-domain (FDTD) approach is adopted to calculate the absorption performance of the HRA. The reflection of the HRA is defined by the ratio of the reflected power to the launched power. Periodic boundary conditions are used in the *x* direction, and perfectly matched layer boundary conditions are used in the *z* direction. The grid size is chosen as 2 nm in both the *x* and *z* directions. In practice, the HRA can be fabricated by using the standard nanolithography processes [3,32,33], as shown in Figure 1b. First, a 100-nm-thick Ag layer is deposited on top of the K9 glass. Next, a thick positive photoresist is spin-coated onto the Ag layer. Photolithography is used to create the rectangular array by double-beam interference lithography with the designed exposure process. Then, an electron beam evaporation system is used to deposit the alternating 10 nm Ag and 35 nm SiO$_2$ multilayer. Finally, the HRA can be obtained after the sample is soaked in acetone to lift-off the photoresist layer.

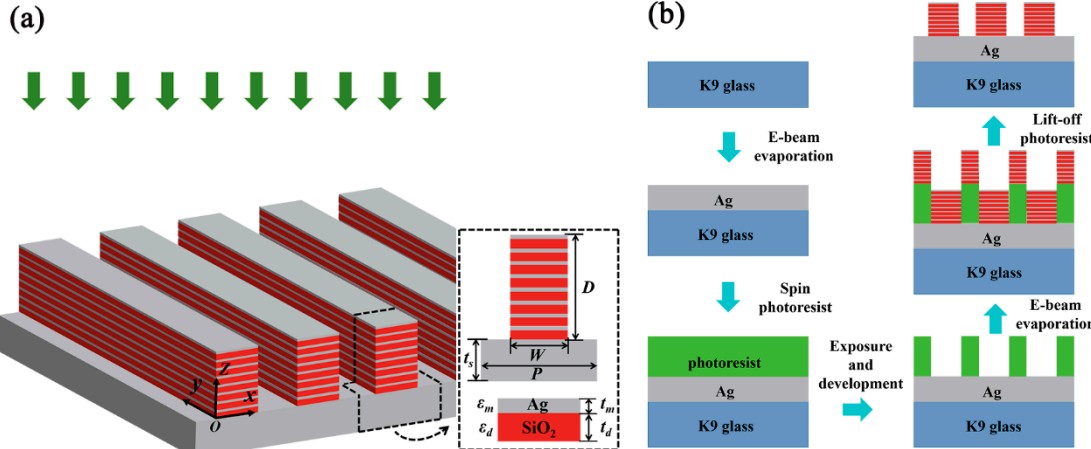

**Figure 1.** (**a**) Schematic diagram of the HRA illuminated by normal incident TM wave (magnetic field vector lies along the *y*-axis). The parameters are: $P = 500$ nm, $W = 375$ nm, $D = 360$ nm, $t_m = 10$ nm, $t_d = 35$ nm, $f = 0.22$, and $t_s = 100$ nm. (**b**) Fabrication process flow of the HRA.

The structure of alternating Ag/SiO$_2$ pairs can be represented as the ultra-anisotropic limit of a traditional uniaxial crystal, and the corresponding dielectric permittivity tensor can be expressed as [4]:

$$\varepsilon = \begin{pmatrix} \varepsilon_\perp & 0 & 0 \\ 0 & \varepsilon_\perp & 0 \\ 0 & 0 & \varepsilon_\| \end{pmatrix} \tag{1}$$

Here, $\perp$ and $\|$ indicate components perpendicular and parallel to the Ag/SiO$_2$ interfaces, respectively. Since the thickness of a film stack ($t_m + t_d$) is much smaller than the wavelength, the dielectric constant of the HRA can be estimated using the EMT. According to the EMT of HMM [34], the perpendicular and parallel components of dielectric permittivity can be written as:

$$\varepsilon_\perp = f\varepsilon_m(\omega) + (1-f)\varepsilon_d \tag{2}$$

$$1/\varepsilon_\| = [f/\varepsilon_m(\omega)] + [(1-f)/\varepsilon_d] \tag{3}$$

For the HRA illuminated by a TM-polarized pane wave, its dispersion relation can be determined by:

$$\frac{k_x^2 + k_y^2}{\varepsilon_\|} + \frac{k_z^2}{\varepsilon_\perp} = \left(\frac{\varepsilon}{c}\right)^2 \tag{4}$$

where $k_x$, $k_y$ and $k_z$ are the components of the wavevector along the $x$, $y$ and $z$ directions, respectively. $\omega$ is the frequency of the light wave, and $c$ is the speed of light in vacuum. $\varepsilon_\|\varepsilon_\perp < 0$ is required to achieve hyperbolic light dispersion for the HRA structure with alternating Ag/SiO$_2$ pairs.

Figure 2 shows real parts of the perpendicular and parallel components of dielectric permittivity of the HRA calculated using the EMT. The filling ratio of the alternating Ag/SiO$_2$ pairs is $f = 0.22$. As can be seen in Figure 2, the HRA has dielectric permittivity tensor components with opposite signs, indicating that the optical properties of the HRA are related with the hyperbolic dispersion as the wavelength is larger than 500 nm.

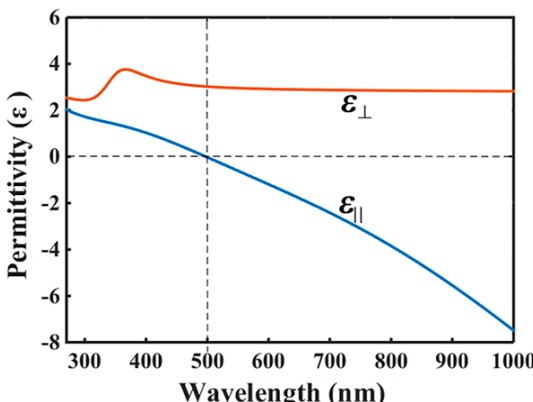

**Figure 2.** Real parts of the $\varepsilon_\perp$ and $\varepsilon_\|$ of dielectric permittivity of the HRA calculated by using the EMT.

## 3. Results and Discussion

Figure 3 shows optical properties of the HRA, and the structural parameters are the same as those in Figure 1a. As can be seen in Figure 3a, dual-band absorption enhancement with a peak efficiency of ~95% are obtained at 712.2 nm and 835.4 nm, which are denoted as peak 1 and peak 2, respectively. The absorption bandwidths of peak 1 and peak 2 are 34.6 nm and 28.0 nm, respectively. Comparing with many previous HMMs-based absorbers, which were mainly designed to achieve broadband absorption enhancement in different wavelength regions [13–27], the absorption response of the HRA shows the property of narrowband absorption enhancement. In addition, the sideband level of the absorption

spectra is low and flat as wavelength is varied. Thus, the HRA possesses good selective absorption performance with double channels.

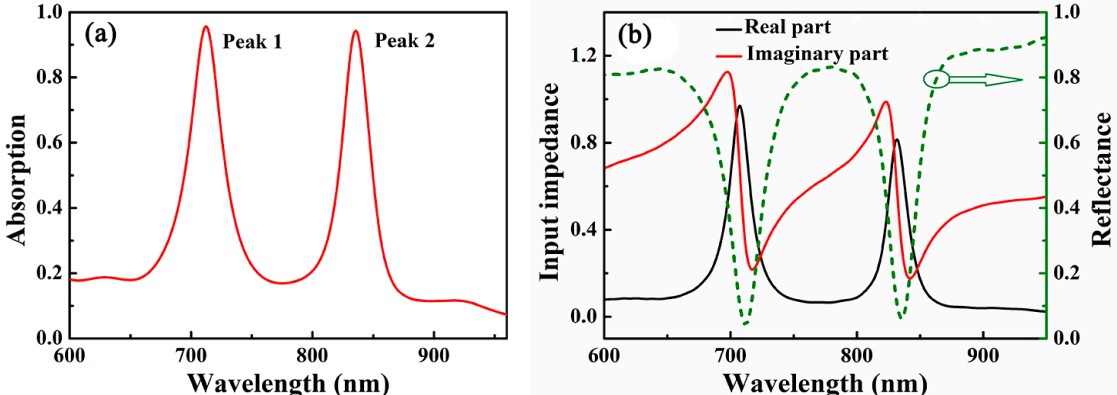

**Figure 3.** Optical properties of the HRA. The parameters are the same as those in Figure 1a. (**a**) Absorption response of the HRA. (**b**) Reflection response and input impedance of the HRA.

To better understand the selective absorption properties of the HRA, the equivalent impedances of the structure are studied by using the impedance theory. According to the impedance theory [35], the input impedance $Z_{in}(\omega)$ and the scattering parameters of the HRA can be generated by:

$$Z_{in}(\omega) = \pm \sqrt{\frac{(1 + S_{11})^2 - S_{21}^2}{(1 - S_{11})^2 - S_{21}^2}} \tag{5}$$

$$s_{21} = s_{12} = \frac{1}{\cos(nkD) - \frac{i}{2}\left(Z_{in}(\omega) + \frac{1}{2}\right)\sin(nkD)} \tag{6}$$

$$s_{11} = s_{22} = \frac{i}{2}\left(\frac{1}{Z_{in}(\omega)} - Z_{in}(\omega)\right)\sin(nkD) \tag{7}$$

where $S_{11}$, $S_{21}$, $S_{12}$, $S_{22}$ are scattering parameters; $k$ is the wavevector, and $n$ is the effective refractive index of the HRA. Therefore, the reflection of the HRA can be calculated as:

$$R = \left[\frac{Z_{in}(\omega) - Z_0}{Z_{in}(\omega) + Z_0}\right]^2 \tag{8}$$

where $Z_0$ is the impedance of the free space, and $Z_0 = \sqrt{\mu(\omega)/\varepsilon(\omega)} = 1$. Obviously, to reduce the reflection of the HRA, the input impedance $Z_{in}(\omega)$ should be matched with the impedance of the free space $Z_0$ for both two absorption bands.

Figure 3b shows reflection response and input impedance of the HRA. As can be seen in Figure 3b, both the real and imaginary parts of $Z_{in}(\omega)$ are varied abruptly around the absorption bands so as to meet the impedance matching condition. In particular, the real part of $Z_{in}(\omega)$ approaches 1, and the imaginary part of $Z_{in}(\omega)$ tends to 0 at the wavelengths of peak 1 and peak 2, which confirms the preceding theoretical analysis on dual-band absorption enhancement of the HRA.

In order to evaluate the validity of the EMT for estimating absorption performance of the HRA, the absorption spectra of the HRA and the equivalent structure are shown in Figure 4, where the equivalent structure is shown in the figure inset. As can be seen in Figure 4, the absorption response calculated using the EMT shows a similar curve to the HRA. The absorption peaks of the EMT are slightly deviated from those of the HRA. This is mainly because the number of film stacks is assumed to be infinite ($N = \infty$) in the EMT approximation. However, the properties of dual-band absorption enhancement of the HRA can be well demonstrated using the EMT.

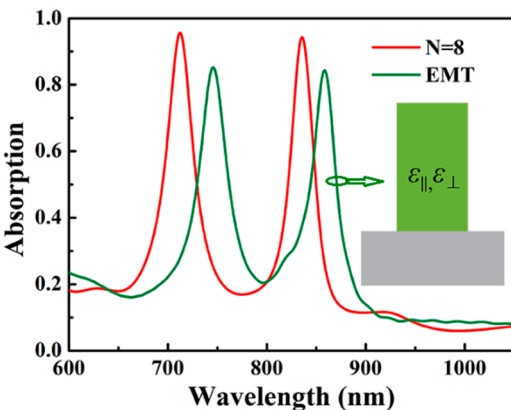

**Figure 4.** Absorption response of the HRA with parameters the same as those in Figure 1a, and absorption response of the equivalent structure.

To investigate the physical origin of the dual-band light absorption enhancement of the HRA, near field distributions of the $E_z$ and $H_y$ components at peak 1 and peak 2 are calculated. Figure 5 shows field distributions of the HRA associated with the two absorption peaks. As can be seen in Figure 5a,b, surface plasmon resonance (SPR) is excited at the interface of the film stack, and the electric-field energy is mainly confined in the HRA. In Figure 5c,d, the magnetic field is also greatly enhanced, and the magnetic-field energy is trapped by the HRA as well. The SPR shows a slightly higher electric-field distribution at 835.4 nm compared with 712.2 nm, thus the absorption bandwidth of peak 2 is smaller than that of peak 1. By observing the magnetic-field distribution of the HRA, the absorption enhancement at peak 1 and peak 2 can be identified as second-order and first-order PSP modes, respectively [36].

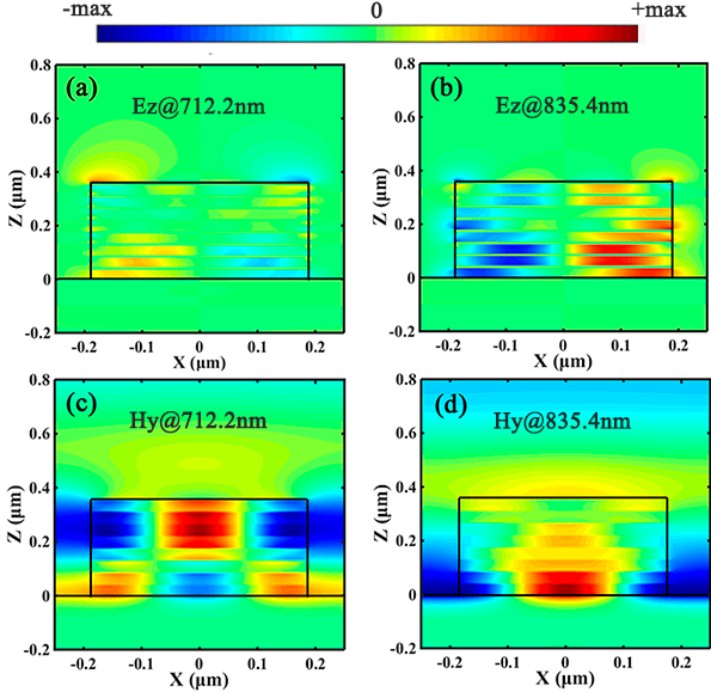

**Figure 5.** Field distributions of the HRA associated with the absorption peaks. (**a**,**b**) are the near field distributions of $E_z$ at 712.2 nm and 835.4 nm, respectively. (**c**,**d**) are the near field distributions of $H_y$ at 712.2 nm and 835.4 nm, respectively.

The influence of the width and the number of film stack on absorption response of the HRA is also studied. As can be seen in Figure 6, the dual-band absorption enhancement of the HRA is very

robust to the variations of $W$ and $N$. In Figure 6a, it shows that both the two absorption peaks shift to the longer wavelength as $W$ is increased. The increase of $W$ increases the length of the anisotropic metamaterial waveguide, resulting in the redshift of the absorption peak of the HRA [37]. Figure 6b shows absorption response of the HRA as a function of $N$ with the fixed $f$ and $D$, and the absorption response of the HRA is calculated by using the EMT. As seen in Figure 6b, the absorption response of the HRA approaches that of the EMT as $N$ is increased, and it almost coincides with that of the EMT as $N = 64$. In Figure 6c,d, it can be seen that the $\lambda_{peak}$ difference between the HRA and the EMT is reduced as $N$ is increased, indicating that the accuracy of the EMT can be improved by increasing the number of film stack.

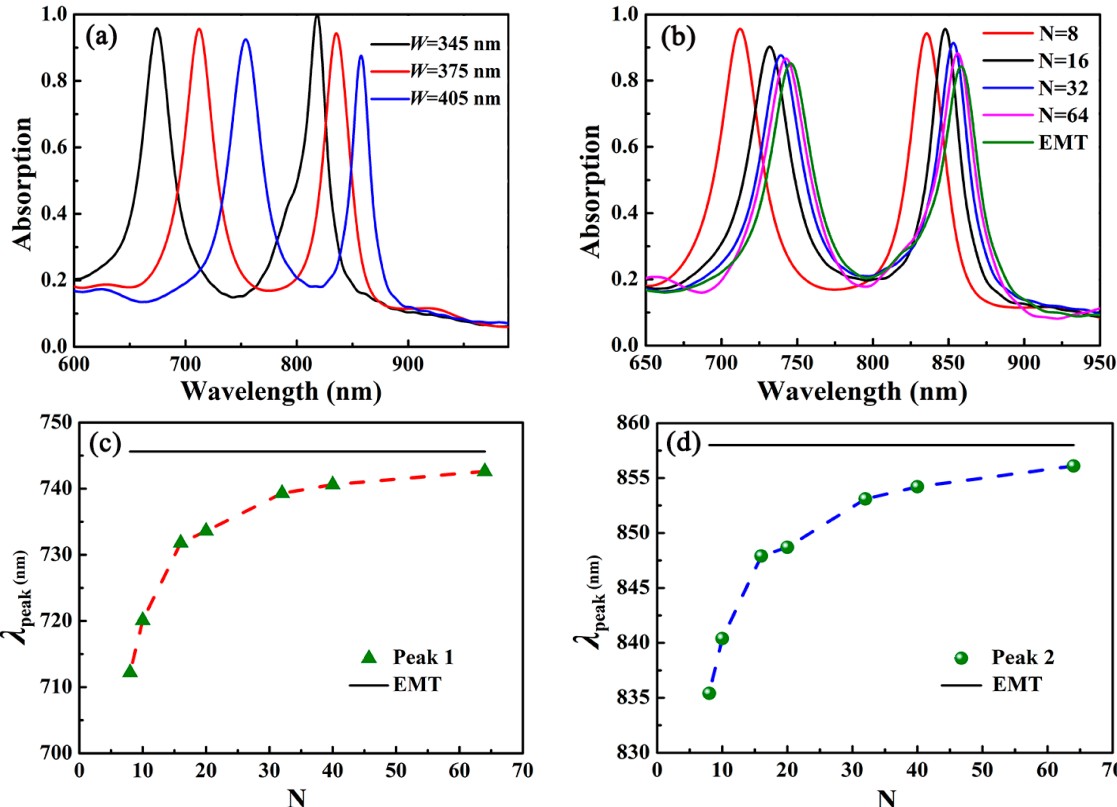

**Figure 6.** Influence of the width and the number of film stack on absorption response of the HRA. Other parameters are the same as those in Figure 1a. (**a**) Absorption response of the HRA as a function of $W$. (**b**) Absorption response of the HRA as a function of $N$ with the fixed $D$ and $f$; absorption response of the HRA calculated by using the EMT. (**c**) Location of peak 1 for different $N$ comparing with that of the EMT. (**d**) Location of peak 2 for different $N$ comparing with that of the EMT.

Finally, we showed that the HRA can be functioned as the refractive index sensor due to the robust performance of dual-band absorption. In general, various techniques have been investigated for sensing a special change of the refractive index. These techniques include nonlinear plasmonic sensing [38], guided-mode resonance [39], Mach-Zehnder interferometers [40], and whispering gallery mode resonators [41]. Here, we evaluate the refractive sensing capabilities of the HRA with sensitivity $S = \Delta\lambda_{peak}/\Delta n$ and figure of merit (FoM) = $S/FWHM$, where $\Delta\lambda_{peak}$ is the peak wavelength change with the refractive index change $\Delta n$, and $FWHM$ is the half-width of the absorption band [42]. As can be seen in Figure 7a, both the two absorption peaks of the HRA redshift as the refractive index of the background is increased, and the low sideband level of absorption response can be kept almost the same. As shown in Figure 7b,c, the slope of the curves in the figures show that the sensitivity reaches 200 nm/RIU and 101 nm/RIU for peak 1 and peak 2, respectively. The sensitivity of peak 1 is larger, although the SPR of peak 1 shows a slightly lower field distribution. This may be because the field

energy of peak 1 is less localized by comparison with peak 2, and more field energy of peak 1 penetrates into the background, resulting in larger sensitivity. The FoM for peak 1 and peak 2 are 5.8 and 3.6, respectively. The FoM of the HRA is relatively small due to the large *FWHM*. However, the sensitivity of the HRA is comparable with many other nanostructures, such as waveguide gratings [43–45] and plasmonic absorbers [46,47] in the visible and near-infrared region, and there is a good linearity between the absorption peak and the refractive index of the background. Therefore, dual-band absorption enhancement of the HRA may be suitable for sensing-related applications.

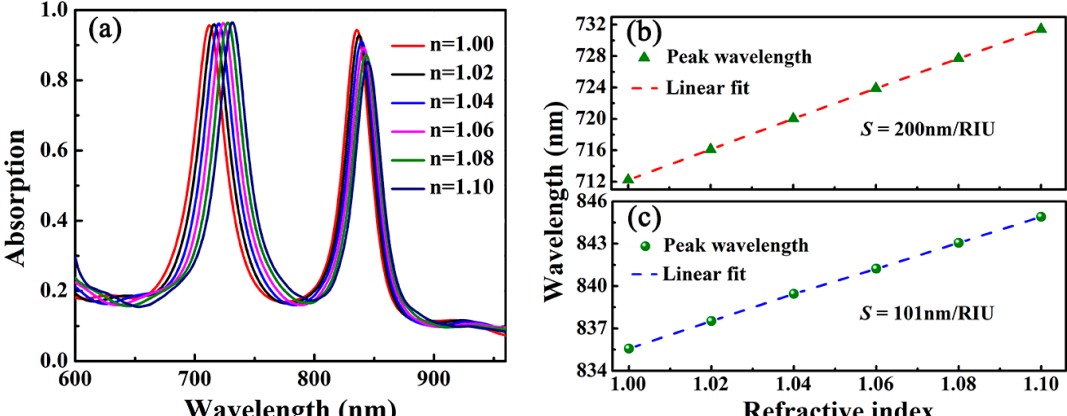

**Figure 7.** Sensor performance by utilizing dual-band absorption of the HRA. Other parameters are the same as those of Figure 1a. (**a**) Absorption response as a function of refractive index of the background. (**b**) Location of peak 1 as a function of refractive index of the background. (**c**) Location of peak 2 as a function of refractive index of the background.

## 4. Conclusions

Dual-band light absorption enhancement in the HRA is presented, and double absorption bands with peak absorption of ~95% in the visible-to-near-infrared region are obtained. The enhanced light absorption of the HRA resulted from the PSP resonance, and the absorption channels at short and long wavelengths are identified as the second-order and first-order PSP modes, respectively. The impedance theory is used to evaluate the absorption properties of the HRA, which shows that both the real and imaginary parts of input impedances are varied abruptly around the absorption bands so as to satisfy the impedance matching. The absorption spectra of the HRA can be estimated by using the EMT, and the accuracy of the EMT can be improved as the number of film stacks is increased. The dual-band absorption is very robust to the variations of the width and the number of film stack, and good refractive index sensing performances can be achieved by using the two absorption bands. The proposed strategy of the HRA can, in principle, be applied to other HMMs systems and could be useful in diverse applications, including thermal emitters, photovoltaics and multi-analyte sensors.

**Author Contributions:** In this work, H.Q. and T.S. performed the design, analyzed the data, and drafted the manuscript; L.W. and X.Y. performed numerical simulations; J.W. and Y.W. guided the idea and checked the figures. All authors read and approved the final manuscript.

**Funding:** This research was funded by the National Natural Science Foundation of China (Grant No. 11811530052), Fundamental Research Funds for the Central Universities (Grant No. JUSRP21935), and Jiangsu Provincial Research Center of Light Industrial Optoelectronic Engineering and Technology (Grant No. BM2014402).

**Conflicts of Interest:** The authors declare no conflict of interest.

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
