# Peer review of "Dual-Band Light Absorption Enhancement in Hyperbolic Rectangular Array"

_applsci, doi:10.3390/app9102011_

Round 1

Reviewer 1 Report

Overall, the investigation is interesting and demonstrates nicely the potential of dual-band light absorber based on a hyperbolic rectangular array for sensing. The introduction contains a good state of the art. I would recommend to move the paragraph about effective medium theory in the Design Principles section, so to present the hyperbolic optical regime at an earlier stage. I would add a more quantitative comparison with state-of-the-art absorbers. Finally, I would suggest adding a paragraph on fabrication guidelines. Giving that these minor changes along with others reported below will be carried out, I recommend the manuscript for publication in AS. The work numerically and analytically proves hyperbolic rectangular array dual-band absorber as a promising platform in the field of sensing

Some additional comments:

Pg 3 Line 89 

Add a table of comparison with state-of-the-art absorbers.

Pg 4 Line 108-126

I would move this paragraph in the Design Principles section, so to present the hyperbolic optical regime at an earlier stage.

Pg 5 Fig.4

The two resonances show hot spots at the interface of the film stack, with the first order propagating surface plasmon showing slightly higher field distribution. This results in a narrower peak distribution absorbance peak. Is this responsible for an increased sensitivity with respect to the other peak? Can you comment on that? 

Author Response

Dear Reviewer,

Thank you very much for your careful review of our manuscript (Title: Dual-Band Light Absorption Enhancement in Hyperbolic Rectangular Array; Manuscript ID: applsci-494395).

We have carefully revised our manuscript based on your constructive comments, and all the revised places are written in pink so that you can check them easily. The detailed contents of revision and revised paper are listed PDF file below.

Reviewer 2 Report

Review of " Dual-Band Light Absorption Enhancement in Hyperbolic Rectangular Array " by Honglong Qi et al.  

This paper reports the investigation of dual-band light absorption enhancement in hyperbolic rectangular array. I believe that the study could potentially be valuable to the optical field due to the opportunity to design the absorption curve which could be utilized in sensing applications. However, I can only recommend this paper to be published in Applied Sciences after major revisions.  My main criticism is summarized below: 

1) It would be important to explain sensing application results in the context of the previous research. In particular, advantages/disadvantages of the suggested approach over other methods (e.g. using optical whispering galley mode resonators). Appropriate references and brief discussion is necessary to make this work more relevant to the readers. 

2) It is unclear from the manuscript how material’s parameters in the presented theoretical study are relevant to the current fabrication capabilities. 

3) It is important to provide the detailed description on the methods used to obtain Fig. 4, Fig. 6, Fig. 3a. 

4) In general, I found that manuscript is very poorly written. Spelling and writing style needs to be improved. The Figure 4 was not discussed in the manuscript at all! 

Author Response

(The authors gave the same response as above.)

Round 2

Reviewer 2 Report

Authors provided sufficient response to my previous comments.